# Investigate Non-EPI Vaccination Recommendation Practice from a Socio-Ecological Perspective: A Mixed-Methods Study in China

**DOI:** 10.3390/vaccines10122105

**Published:** 2022-12-08

**Authors:** Kaiyi Han, Zhiyuan Hou, Shiyi Tu, Qian Wang, Binbing Wang, Xiaoyu Liu, Shiqiang Jiang, Tracey Chantler, Heidi Larson

**Affiliations:** 1School of Public Health, Fudan University, Shanghai 200032, China; 2Department of Infectious Disease Epidemiology, London School of Hygiene & Tropical Medicine, London WC1E 7HT, UK; 3NHC Key Laboratory of Health Technology Assessment, Fudan University, Shanghai 200032, China; 4Anhui Center for Disease Control and Prevention, Hefei 230061, China; 5Shaanxi Center for Disease Control and Prevention, Xi’an 710054, China; 6Nanshan Center for Disease Control and Prevention, Shenzhen 518054, China; 7Department of Global Health and Development, London School of Hygiene & Tropical Medicine, London WC1E 7HT, UK; 8Department of Health Metrics Sciences, University of Washington, Seattle, WA 98195, USA

**Keywords:** vaccination, non-EPI vaccine, recommendation, communication, health care worker, China

## Abstract

The uptake of non-EPI vaccines, such as influenza and pneumonia vaccines, are very low in China compared to other countries. In China, immunization services are provided by dedicated vaccination service providers (VSPs), and their recommendation is the key to improve vaccine uptake. This study explores VSP recommendation practices for non-EPI vaccines from a socio-ecological perspective. A mixed-methods study, combining a questionnaire survey and key informant interviews, was conducted in Anhui, Shaanxi, and Guangdong provinces. 555 VSPs completed the valid questionnaire, and 49 VSPs participated in in-depth interviews. Among the surveyed VSPs, 51.54% stated that they always or often recommended non-EPI vaccines in work, and the remaining half reported that they sometimes or never recommended non-EPI vaccines. Most VSPs interviewed communicated about non-EPI vaccines with the public in an informed style, not a presumptive one, and provided the public with all the decision-making latitude. The infrequent recommendation of non-EPI vaccines was widely prevalent among Chinese VSPs regardless of their individual characteristics, and was mainly driven by the interpersonal relationship, institutional arrangement, and public policy. Firstly, the VSPs were concerned about conflicts arising from the recommendation of self-paid vaccines and the risk of adverse reactions following vaccination. Secondly, high workloads left them insufficient time to communicate about non-EPI vaccines. Thirdly, there was no performance assessment or financial incentive for VSPs to recommend non-EPI vaccination, and their main responsibility was around EPI vaccination. Therefore, multi-level socio-ecological systems around non-EPI vaccination should be improved to optimize the communication between VSPs and the public, which include a better system of legal redress to resolve potential misunderstandings between the VSPs and the public, more effective workload management through whole-process health information system and strengthening public health workforce, and the introduction of performance assessment and appropriate incentives on non-EPI vaccination.

## 1. Introduction

Immunization has proven to be one of the most cost-effective health interventions [1,2]. China initiated its national expanded program on immunization (EPI) in 1978. Currently, the EPI program includes 14 vaccines against 15 vaccine-preventable diseases, which are provided to children free of charge and are required for school enrolment [3]. Vaccines not covered by the EPI program can be accessed voluntarily but must be paid for (Appendix A). Compared to almost universal coverage of EPI vaccines, the uptake rates for non-EPI vaccines remain low in China [4,5,6]. For example, according to a survey, influenza vaccine (a non-EPI vaccine) uptake for young children was only 3.1% during the 2014–2015 influenza season in Xiamen city [7]. Research conducted in three provinces, with different socio-economic characteristics in China in 2013 suggests that vaccine affordability could explain this low uptake; Hou et al. found that the majority of caregivers of children zero–three years old were not willing to pay the market price for non-EPI vaccines [8].

While the reasons for the low uptake of non-EPI vaccines are complex, healthcare workers (HCWs) can play a core role in supporting public confidence in vaccination and making vaccination more accessible. There is a significant body of evidence that suggests that HCWs are the most trustworthy sources of health information for the public [9,10] and communication between HCWs and the public is considered to be the cornerstone of maintaining the public’s confidence in vaccinations [11]. In many countries, such as United States and United Kingdom, vaccination services are provided by HCWs, such as general practitioners and nurses, who also provide other medical services to the public [12]. In China, however, immunization services are provided by dedicated vaccination service providers (VSPs) at vaccination clinics held in community health centers [10]. Vaccinators are fully trained to deliver immunization programs and schedule appointments with the public directly. A community health center in China typically employs 40–100 HCWs, of which only three–five serve as vaccinators, and generally HCWs do not have any vaccination responsibilities.

Two systematic reviews have summarized the determinants of HCWs’ recommending the HPV vaccine worldwide, and found that recommendation behaviors varied by HCWs’ knowledge, perceptions, and professional characteristics [13,14]. Very few studies have investigated Chinese HCWs’ vaccine recommendation behaviors. Previous studies reported that a low proportion (56.26%) of HCWs recommended the influenza vaccination to children in China, and that public health workers were more likely to recommend flu vaccine in contrast to general practitioners, as were those who had received a flu vaccination and those with more knowledge about national influenza vaccination guidelines [15,16].

There are limitations with the existing studies that investigate HCW-patient communication and recommendation for vaccines. Firstly, HCW-patient communication can be rendered in three styles: Informed, shared, or presumptive [17]. These three different styles vary in the flow of information exchange, the leading role in expressing treatment preferences, and choosing a treatment to implement, and therefore have different levels of strength of recommendation for vaccines. In the informed and presumptive styles, the information exchange is largely one way and the HCW is assumed to be the primary source of information to the patient on medical issues about the patient’s disease and treatment options, however, in the former, the HCW has no further role in the decision-making process, in the latter, the treating HCW may communicate to the patient only the ultimate treatment decision, failing to reveal knowledge and values considered in the selection process and how these were weighted. In the shared style, the information exchange is two-way, and both sides work towards reaching an agreement and have an investment in the ultimate decision made. Presumptive style communication from HCWs, a more HCW-driven communication style, has been associated with decreased hesitancy and increased receipt of vaccination [18]. Most existing studies do not take these communication styles into account. Secondly, individual behavior is viewed as being affected by multiple levels of familial, social and cultural influences. The WHO Strategic Advisory Group of Experts on Immunization developed a determinants matrix for vaccine hesitancy, which covers contextual influences, individual and group influences, and vaccine/vaccination-specific influences [19]. Previous studies on vaccination recommendation mainly focus on the intrapersonal layer such as HCWs’ knowledge and perception but lack an overarching framework that incorporates the influences from other layers, such as institution regulation and policy.

The social-ecological model provides a conceptual framework to direct attention to both behavior and its individual and environmental determinants [20,21]. This model presents behavior as a product of the interdependence between the individual and subsystems of the ecosystem (e.g., family, community, culture, physical and social environment) [20]. In this model, patterned behavior is the outcome of interest and is viewed as being determined by five sub-ecosystems, which are intrapersonal, interpersonal, institutional, community, and policy. It has been used as a framework for studying medical services, such as non-prescription antibiotic dispensing [22]. It can also help to investigate HCWs’ recommendation behaviors for vaccines in a comprehensive manner. This study aims to frame the potential determinants of HCWs’ recommendation for non-EPI vaccines in China from a socio-ecological perspective. A mixed-methods study combining a cross-sectional survey and key informant interviews was adopted for this purpose. Our target population was VSPs since they are dedicated to deliver vaccination services in China, instead of general HCWs.

## 2. Materials and Methods

### 2.1. Study Design

We conducted a mixed-methods cross-sectional study in January 2019 in Shenzhen megacity, Anhui province, and Shaanxi province, covering the East, Middle and West of China, respectively. One urban district and one rural county were selected separately in the Anhui and Shaanxi provinces, and one urban district was selected in Shenzhen megacity. In total, this study was conducted in five districts/counties in China.

The Fudan University School of Public Health, and the London School of Hygiene & Tropical Medicine Ethics committees approved the study protocol [FDU IRB#2018-10-0703, LSHTM Ethics Ref 16016].

### 2.2. Data Collection

#### 2.2.1. Survey of Vaccination Service Providers

To estimate the recommendation practice of non-EPI vaccines, a cross-sectional survey was conducted for all VSPs in the sampled districts and counties. A multi-stage sampling process was used to ensure the representativeness of the sample. Guangdong Province, Anhui Province and Shaanxi Province were selected to represent higher, median and lower social-economic tiers, respectively. At the provincial level, one urban district and one rural county were included in Shaanxi and Anhui provinces, and one urban district was included in Shenzhen megacity, Guangdong province. All VSPs (600) working in the sampled areas were invited to participate in a mobile-phone-based questionnaire survey by scanning a QR code. The self-administered questionnaire was distributed and managed using the online platform Wenjuanxing (https://www.wjx.cn/ (accessed on 9 June 2021)).

#### 2.2.2. Interview

To understand the determinants of recommendation practice of non-EPI vaccines in depth, semi-structured interview was conducted following the questionnaire survey. In each sampled district/county, we interviewed one immunization program manager from CDC, and VSPs from vaccination clinics in three selected community health centers. These three community health centers were selected to represent low, medium, and high socio-economic tiers within each district/county. Generally, there are 3–5 VSPs at a vaccination clinic, who are the director in charge of the clinic, vaccinators for vaccination service delivery and consultation, and a pediatrician for medical pre-screening and adverse reaction response. In each vaccination clinic, we invited one VSP from each job responsibility to participate in an interview.

### 2.3. Instruments

#### 2.3.1. Questionnaire

The questionnaire was piloted for 10 VSPs in two non-study communities in Shanghai (Appendix B). The content of the questionnaire included the (a) study site, rural or urban residence, gender, age, education level, and profession (doctors, nurses or public health workers); (b) recommendation frequency of non-EPI vaccines, measured using the following question—How often do you recommend non-EPI vaccines to the public?. There were four response options—“always”, “often”, “sometimes” and “never”. Response options were further grouped into two categories for the analysis: often (including “always” and “often”), and not often (including “sometimes” and “never”). The question were linked to previous studies in the fields of HCWs’ recommendation practice of vaccines [16].

#### 2.3.2. Interview Guides

We developed interview guides according to five sub-ecosystems of the social-ecological model (Appendix C) [20]. First, we asked interviewees about their communication and recommendation of vaccines to the public in their daily work. In terms of intrapersonal sub-ecosystem, we focused on VSPs’ knowledge, perception, and confidence in vaccines and vaccination services. For interpersonal sub-ecosystem, we asked about the quality of doctor-patient relationships and relationships with other colleagues. For the institutional sub-ecosystem, we asked the VSPs about their routine work, self-evaluation of workload, and the potential impact of both on the recommendation practice of non-EPI vaccines. For community sub-ecosystem, we enquired about the supply of non-EPI vaccines and whether any shortage of non-EPI vaccines had ever occurred. For public policy sub-ecosystem, we investigated the influence of financial incentive policy on the recommendation practice of non-EPI vaccines and assessment from superiors (CDC).

All participants were informed of the purpose of the study. They were also informed that participation was voluntary and that they could withdraw at any time. All participants were assured of the confidentiality of the interviews. Each interview lasted between 30 and 60 min and were audio-recorded after obtaining written informed consent. 

### 2.4. Statistical Analysis

#### 2.4.1. Statistical Analysis for Survey Data

The recommendation practice of non-EPI vaccines was measured by the proportion of VSPs, who often recommend non-EPI vaccines among the total sample. Univariate analyses were performed to compare the VSPs’ recommendation practice of non-EPI vaccines by their socio-demographic characteristics using Chi-square tests. A multivariable logistic regression analysis was further conducted to examine the factors associated with the VSPs’ recommendation practice of non-EPI vaccines. Odds ratios with 95% confidence intervals were presented. All survey data were analyzed using STATA, version 14.0 (Stata Corp, College Station, TX, USA).

#### 2.4.2. Data Analysis for Interviews

All interviews were transcribed verbatim and checked by another investigator. We conducted a thematic analysis using a combination of deductive and inductive coding to analyze the transcripts of the interviews [23]. We first identified detailed sub-themes via deductive, iterative coding of the data. Subsequently, exemplary data extracts were selected from the key sub-themes for inclusion as quotations. The interview transcripts were independently coded by two investigators, and any discrepancies were then discussed until a consensus was reached. All qualitative analysis were conducted using NVivo, version 11 (QSR International Inc., Burlington, MA, USA).

## 3. Results

### 3.1. Quantitative Results

Surveyed VSPs’ characteristics and recommendation practices for non-EPI vaccines are summarized in Table 1. Respondents who completed the questionnaire in less than 2 min or left more than 50% of the questionnaire incomplete, 45 in all, were excluded. In total, 555 of 600 VSPs completed the valid questionnaire. Of the 555 respondents, 15.32% and 36.22% stated that they always or often recommended non-EPI vaccines to patients in work, whereas 36.4% and 12.07% of respondents reported that they sometimes or never recommended these vaccines, respectively.

Results from multivariate logistic regression (Table 1) suggested that respondents living in Anhui province were significantly more likely to recommend non-EPI vaccines than those in Shenzhen city (OR = 1.52, 95%CI: 1.04–2.20). VSPs older than 45 years old were significantly more likely to recommend non-EPI vaccines than those younger than 25 years old (OR = 2.50, 95%CI: 1.42–4.39). However, rural or urban residence, gender, education level, and professions had no significant association with recommendation practices for non-EPI vaccines.

### 3.2. Qualitative Results on Health Education and Recommendation Practices for Non-EPI Vaccines

In total, we conducted 43 interviews with VSPs and six interviews with immunization program managers (Table 2). Communication about non-EPI vaccines between VSPs and the public covers health education and recommendation practices.

#### 3.2.1. Health Education on Immunization

Most participants said that health education on immunization (including education to parents of newborn babies) was provided routinely in their workplace. The content mainly covered the importance of vaccination and the introduction of the EPI in China. As one VSP noted: 

“First, we will give a general explanation of the components of the vaccine. Then, patients could wonder, some vaccines are free, and the other are not, why? Any difference between those two types of vaccines? We will tell, every vaccine is of the same importance. We also want parents to make sufficient preparation before vaccination. We will tell them to focus on five things: wearing the right clothes to keep warm; […]. We need to popularize these for parents. The main thing is to get them to understand the importance of vaccination, the safety, right? And vaccines are very cost-effective.” (VSP 4, male, Dongzhi county, Anhui province).

#### 3.2.2. Recommendation Practices for Non-EPI Vaccines

Most VSPs said that they informed parents about age-appropriate vaccines for their children and asked about their intention to be vaccinated (mainly non-EPI vaccines) after the completion of EPI vaccinations. However, they did not actively recommend non-EPI vaccines. Almost all VSPs said that the purpose of this notification was to remind parents of the availability of non-EPI vaccines, and at the same time, honor parents’ decision-making autonomy on non-EPI vaccinations for their children. As one VSP noted:

“Definitely no recommendation, but every time after finishing one free (EPI) vaccination, I would talk to them. It’s like, before the next free (EPI) vaccine, there are other vaccines available, they are voluntary and not free. Then I would tell them, if you want to get it, I can make another appointment for you. If you don’t, we won’t force you to get vaccinated. It’s voluntary, basically. They would ask, didn’t you say vaccination was free? Then I say this is non-EPI vaccine, you can choose to get it or not […].” (VSP 3, female, Dongzhi county, Anhui province).

### 3.3. Qualitative Results on the Ecosystems Influencing Recommendation Practices for Non-EPI Vaccines

#### 3.3.1. Intrapersonal Sub-Ecosystem

Participants expressed the high confidence in vaccines and vaccination services no matter which are covered by the EPI or not. They believed that the benefits of vaccination outweigh the risks in general. As one participant said:

“I think I agree with the statement (the benefits of vaccination outweigh the risks), it’s not because I work on this […]. Vaccines like influenza, my colleague’s child got influenza vaccination, and then went to kindergarten. There are more than 40 children in the class, and only a dozen of them can come to school this time (Others stayed away from school because they had the flu). But his child has been fine and have not caught the flu.” (VSP 27, female, Shushan District, Anhui province).

Meanwhile, participants indicated a lack of knowledge about vaccines. They knew the vaccination schedule and service procedure, but did not know about data on the effectiveness or safety of specific vaccines. As one participant said:

“I think it’s… Just my knowledge about these vaccines… is too little, I know too little about it […]. Most parents don’t ask too much, but we really know little… my knowledge isn’t very comprehensive.” (VSP 41, female, Jingyang County, Shaanxi province).

#### 3.3.2. Interpersonal Sub-Ecosystem

Many VSPs indicated that they were concerned about adverse reactions following vaccination, which could cause conflicts between parents and themselves if they recommended non-EPI vaccines to parents. As one participant said:

“One problem is. In one hospital, there was a case of adverse reaction related to non-EPI vaccination, and the dispute is very tricky. I remember that they compensated for it. They (the hospital) make so little money on vaccination, but finally have to pay so much compensation. They can’t even carry out routine work at that time. Later, because of this, they almost gave up the inoculation of non-EPI vaccine. We just don’t want to do it. The dean thought this was so tricky and he did not want to get involved in non-EPI vaccine. This case really hit him hard.” (VSP 39, female, Jingyang County, Shaanxi province).

Some VSPs also stated that parents resent being recommended paid medical services (including vaccines). Therefore, recommending non-EPI vaccines may lead parents to perceive that the healthcare providers are profit-seeking and may further reduce parents’ trust in them. As one participant said:

“We don’t recommend it, only inform them (with the age-appropriate vaccines). Why? They will be unsatisfied. For example, we will tell him that there are two kinds of Hepatitis A vaccines, one is imported, the other is domestic, and we let parents choose on their own. They would ask which one is better? Go online for information, we just tell you we have the vaccine.” (VSP 15, female, Nanshan District, Shenzhen city).

#### 3.3.3. Institutional Sub-Ecosystem

Many VSPs said that heavy workloads leave them insufficient time to communicate to parents about vaccines. As one VSP said:

“[…]., I need to vaccinate more than 100 people a day. I remember a training I received before, it goes like, vaccination service provider should not vaccinate more than 50 people per day, otherwise, his/her working status will be negatively affected, and he/she may make mistakes, or not be able to communication well with parents, so the satisfaction of parents will decrease […].” (VSP 26, female, Shushan District, Anhui province).

In addition to the heavy workload of vaccinating itself, vaccinators often mentioned two other reasons contributing to their increased workload. Firstly, since many vaccination clinics are not equipped with electronic information system, all the work, including the reminders for children’s vaccination appointments and entry of vaccination information, needs to be done manually. Secondly, some vaccinators said that, in addition to their vaccination work, they are also given other public health responsibilities within their respective jurisdictions, such as a health check-up. As two participants said:

“There is too much work to serve so many people. Now the requirements are so strict, and more and more detailed, right? Registration work, for example, can take you a whole morning if you write it by hand. If there is a set of electronic information system, first, it could alleviate the shortage of workforce, then avoid some mistakes […].” (VSP 4, female, Dongzhi County, Anhui province).

“What I’ve been thinking is how to fulfill the annual work plan, I think a lot, but the plan just couldn’t catch up with change. Our VSPs don’t work only on vaccination, but also other types of work, such as poverty alleviation in rural area. Then scheduled work, such as professional improvement, will be disrupted. We also have to carry out physical examination for the elderly every year. It basically takes two months to complete the physical examination for the elderly in the whole town, and we work every day in two months.” (VSP 10, female, Jingyang County, Shaanxi province).

#### 3.3.4. Community Sub-Ecosystem

Many VSPs said that the cost of non-EPI vaccines is too expensive for local residents. High costs make them feel hesitant to recommend it to parents. As a director of a vaccination clinic described: 

“Especially Pentaxim, its price is very high, 500 or 600 Chinese yuan. I do not advocate this vaccine, because we are in rural areas, here residents’ affordability is limited, right? Its demand is not large.” (VSP 1, male, Dongzhi County, Anhui province).

Many VSPs also indicated that there was a shortage of non-EPI vaccines, such as flu vaccine. They said that they could not recommend it to parents if they did not have it in stock. One VSP commented:

“For EPI vaccines, it’s the leprosy vaccine, for non-EPI vaccines, it’s Pentaxim, both vaccines are often out of stock. It was really difficult to conduct vaccination work at that time.” (VSP 17, female, Nanshan District, Shenzhen city).

#### 3.3.5. Public Policy Sub-Ecosystem

All VSPs indicated that the superior unit (District/County CDC) has clear assessment criteria for EPI vaccination rates but not for non-EPI vaccines. A director of a vaccination clinic described it as follows: 

“We will count how many children need to be vaccinated, how many children have been vaccinated. County CDC’s assessment criteria is that the vaccination rate of EPI vaccines should be at least 95%. Depending on the percentage you reach, you reach 80% and you get 80% merit pay, if 90% and then 90% merit pay. There is no assessment for non-EPI vaccines.” (VSP 42, male, Jingyang County, Shaanxi province).

All VSPs said that a small service fee can be charged for non-EPI vaccinations. However, their income was fixed and not related to the number of non-EPI vaccines they administer. Two vaccinators described:

“We are paid a fixed salary. It has nothing to do with the number of non-EPI vaccine used.” (VSP 24, female, Qingdu District, Shaanxi province).

“Non-EPI vaccines have no impact on our performance salary. Our work performance is generally assessed by the dean. It just depends on the working hours… Our performance income has nothing to do with the amount of EPI and non-EPI vaccination services. It’s all arranged by the hospital […].” (VSP 43, female, Jingyang County, Shaanxi province).

## 4. Discussion

This study used a mixed-method design to investigate the patterns and determinants of VSPs’ communication and recommendation for non-EPI vaccines in the Chinese context. Only half (51.54%) of the VSPs often recommended non-EPI vaccines, and the low frequency of recommendation was independent of their individual characteristics. The VSPs routinely conducted health education about vaccination for the public. Most VSPs recommended non-EPI vaccines in an informed style, not a presumptive one, and provided the public with all decision-making latitude. 

Recommendation from HCWs is regarded as one of the most consistent correlates of vaccination [24]. In our study, nearly half of the VSPs never or only sometimes recommended non-EPI vaccines, although they are full-time designated staffs in charge of vaccination services in China. The low level of recommendation practice is consistent with the previous surveys in China [16,25], but much lower than that in US and European countries [26,27,28,29]. Meanwhile, as for the style of VSPs’ communication practice, the qualitative analysis showed that most VSPs did not recommend but instead informed parents about non-EPI childhood vaccines to honor the parents’ decision-making autonomy. That is only information flows from VSPs to parents, but deliberation and decision on a vaccination option are delegated to parents, according to the framework of patient-provider interactions proposed by Charles et al. [17]. It has been shown that provider-driven communication through the shared and presumptive styles was highly effective for encouraging vaccination than the informed style [18]. Therefore, it is necessary to identify the factors associated with VSPs’ communication practice. 

The infrequent recommendation of non-EPI vaccines was widely prevalent among Chinese VSPs in this study sample, no matter their individual characteristics. This indicated that recommendation practices were possibly not influenced by individual characteristics. Among the five sub-ecosystems in the social-ecological model, interpersonal relationship, institutional arrangement, and public policy mainly contribute to the widely infrequent recommendation of non-EPI vaccines in an informed style instead of a presumptive one in China. 

Firstly, at the interpersonal sub-ecosystem, some VSPs were concerned about potential conflicts arising from recommending the self-paid non-EPI vaccines and adverse reactions after administering these vaccines. The recommendation of paid medical services may lead to the patients considering doctors as retailers pursuing profits and reducing their adherence to the doctors’ recommendation [30]. Discontentment from patients and doctors can even lead to the occurrence of adverse events [31]. In China, doctor-patient relationships has deteriorated during the past decade [32]. The tense doctor-patient relationship may be rooted in the Chinese health system with the long history of profit-pursuing medical behaviors and unaffordable medical services before the 2009 healthcare reform [33]. In addition, most parents have the low awareness on vaccine-preventable diseases due to the preventative nature of vaccines, which may make the public more adverse to being recommended vaccines than clinical services. To reduce the concerns of VSPs, it is necessary to address the tense doctor-patient relationship and improve the compensation mechanism for adverse reactions following vaccination. 

Secondly, at the institutional sub-ecosystem, heavy workloads leave VSPs little time to communicate with the public about vaccines. With more vaccines introduced, public demand for vaccination has surged, leading to an inadequate number of VSPs available to meet the demand [34]. Our findings among VSPs who participated in the study in Shenzhen city, for example, were less likely to recommend non-EPI vaccines due to the pressure on their time than those in Shaanxi and Anhui provinces. There is a much higher proportion of young migrant workers and a more developed economy in Shenzhen than the other two provinces [35], which translated to a greater demand for local vaccination services. Moreover, a lack of electronic information systems also contributed to the overload of the VSPs. Dan Gong et al. found that insufficient infrastructure was one of the main barriers of delivering additional vaccines through the national EPI schedule [36]. While most provinces have an immunization information system capable of managing vaccine stocks and keeping official vaccination records, it cannot support vaccination services [37]. For example, vaccination appointment procedures were primarily traditional, using reservation books and oral notification. In addition, VSPs have to take on additional responsibilities, such as chronic disease management [38,39], and this extended work scope has exacerbated the shortage of the VSPs. Therefore, to ensure the reasonable workload of VSPs and leave time for health communications, the government should promote the construction of the whole-process health information system and strengthen the public health workforce through both retaining and recruiting staff, using financial and nonfinancial incentives [36].

Thirdly, at the public policy sub-ecosystem, there were no performance assessments or financial incentives for VSPs to recommend non-EPI vaccination. In China, there is strict performance assessment for EPI vaccination coverage for each vaccination clinic and CDC, and their performance is related to the staff merit pay, meanwhile there is no performance assessment for VSPs regarding non-EPI vaccination as their main responsibility is around EPI vaccination. Therefore, the non-EPI vaccination should also be covered as a part of performance assessment [36]. Moreover, to address the phenomenon of over-prescriptions, China issued the zero mark-up drug policy (including non-EPI vaccines) by disengaging prescribing from profits in 2009 [40]. It was reported that the policy promoted rational use of medicines [41,42]. Unlike drugs such as antibiotics, which are overprescribed and could lead to adverse health consequences, vaccines are preventative and need to be promoted by VSPs through incentives. Previous studies highlighted that HCWs’ recommendation behaviors were notably influenced by financial incentives [43], and the financial incentives were effective in improving the uptake and delivery of health services [44,45,46]. However, implementing financial incentives could bring additional concerns, including neglect of non-incentivized tasks and distorted motivation among HCWs [47]. Thus, to avoid excessive and unnecessary non-EPI provision for economic benefit, it is important to establish an appropriate income distribution system [43], which could balance basic salary and performance-based incentives (e.g., avoiding overly high incentives and overly low basic salary) [48]. In addition to the above measures, governmental engagement can also contribute to the promotion of non-EPI vaccines. Taking rabies vaccine as an example, in order to meet the goal of eliminating dog-mediated rabies by 2030 [49], the Chinese government promoted rabies prevention education programs, particularly in high-risk provinces; meanwhile, the Chinese national reference laboratory for animal rabies provided training to more than 500 laboratory staff from provincial and municipal animal disease control centers [50]. These measures greatly improve the awareness of HCWs and the access to post-exposure prophylaxis, including the rabies vaccine. Since peaking in 2007 with more than 3000 reported human rabies deaths, substantial progress has been made in reducing these deaths [51].

Our study provides important insights into recommendation practices and the different communication styles among VSPs for non-EPI vaccines in China from a socio-ecological perspective. While previous studies have investigated HCWs’ recommendation of influenza vaccines and their intrapersonal determinants (including knowledge and attitudes towards influenza and influenza vaccines) in China [15,16], there has been less attention paid to the influence of macro-level factors, such as demand, system capacity and public policy on VSPs’ recommendation practice. Given that many childhood vaccines are optional and paid out of pocket in low- and middle-income countries [52], the implications of our study could be valuable for China and other countries with similar contexts.

Our study has several limitations. First, the recommendation behavior of VSPs was self-reported and potentially influenced by recall bias. Second, our study only covered three provinces, and our findings may not be generalized to all parts of China. Third, we only interviewed the VSPs who deliver vaccination services, and did not interview general HCWs who are not responsible for vaccination services, but may give health education on vaccination during clinical services. Finally, the study focused on recommendation for non-EPI vaccines in general. Recommendation behaviors may vary across different non-EPI vaccines, and further studies need to consider recommendation for specific non-EPI vaccines.

## 5. Conclusions

Our study reveals a low frequency of VSPs recommending non-EPI vaccines. Fears of potential conflicts with patients over recommending paid medical services, heavy workload, and the lack of performance assessment and financial incentive are the major barriers to VSPs’ recommending practice. The multi-level ecosystem around non-EPI vaccination should be improved to incentivize and support VSPs and the public, which include a better system of legal redress to resolve potential disputes between the VSPs and the public, more effective workload management through the whole-process health information system and strengthening the public health workforce, the introduction of performance assessment and appropriate income distribution system for non-EPI vaccination, and more governmental engagement in infectious disease prevention programs. 

## Figures and Tables

**Table 1 vaccines-10-02105-t001:** Recommendation for non-EPI vaccines and its associated factors among vaccination service providers participating in the survey.

		Recommendation Practices for Non-EPI Vaccines		Univariate Analyses	Multivariate Logistic Regression
	Total, *n* (%)	Always or Often, *n* (%)	Sometimes or Never, *n* (%)	χ2	OR (95% CI)
Location				1.31	
Shenzhen city	132 (23.78)	63 (47.73)	69 (52.27)		ref.
Shaanxi province	250 (45.05)	129 (51.6)	121 (48.4)		1.25 (0.98–1.60)
Anhui province	173 (31.17)	94 (54.34)	79 (45.66)		1.52 (1.04–2.20) *
Residence				0.05	
Rural	126 (22.7)	66 (52.38)	60 (47.62)		ref.
Urban	429 (77.3)	220 (51.28)	209 (48.72)		1.15 (0.76–1.74)
Gender				5.06 *	
Female	483 (87.03)	240 (49.69)	243 (50.31)		ref.
Male	72 (12.97)	46 (63.89)	26 (36.11)		1.12 (0.47–2.69)
Age (years)				10.04 *	
≤25	63 (11.35)	27 (42.86)	36 (57.14)		ref.
25–35	244 (43.96)	117 (47.95)	127 (52.05)		1.14 (0.82–1.58)
35–45	186 (33.51)	100 (53.76)	86 (46.24)		1.40 (0.91–2.15)
>45	62 (11.17)	42 (67.74)	20 (32.26)		2.50 (1.42–4.39) **
Education				0.04	
High school and below	46 (8.29)	24(52.17)	22 (47.83)		ref.
Junior college	237 (42.7)	123(51.9)	114 (48.1)		1.43 (0.67–3.03)
Undergraduate and above	272 (49.01)	139(51.1)	133 (48.9)		1.41 (0.68–2.94)
Profession				9.78 **	
Doctor	99 (17.84)	60 (60.61)	39(39.39)		ref.
Nurse	362 (65.23)	169 (46.69)	193(53.31)		0.65 (0.33–1.27)
Public health worker	94 (16.94)	57 (60.64)	37 (39.36)		1.12 (0.52–2.43)

Notes: Significance level: ** *p* < 0.01, * *p* < 0.05.

**Table 2 vaccines-10-02105-t002:** Characteristics of vaccination service providers participating in the interview.

	Total, *n*
Total	49
County/city	
Rural county, Anhui province	10
Urban city, Anhui province	11
Rural county, Shaanxi province	11
Urban city, Shaanxi province	6
Shenzhen city	11
Position	
Vaccinator	21
Pediatrician	11
Director of vaccination clinics	11
Immunization program director of CDC	6
Gender	
Female	38
Male	11
Age (years)	
≤25	3
25–35	15
35–45	17
>45	14
Education	
High school and below	6
Junior college	19
Undergraduate and above	24

Note: Center for Disease Control and Prevention (CDC).

## Data Availability

The corresponding author had full access to all the data in the study and had final responsibility for the decision to submit for publication.

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
