# Peer review of "Investigate Non-EPI Vaccination Recommendation Practice from a Socio-Ecological Perspective: A Mixed-Methods Study in China"

_vaccines, 2022, doi:10.3390/vaccines10122105_

Round 1

Reviewer 1 Report

Vaccines

Vaccines 2047551

Comments to Authors

Public acceptance of vaccines has become a critical variable for increasing higher vaccination rates. Trust in healthcare providers and communication between providers and the public is important to assess in this process. Your study examines variables that impact patient willingness to receive a number of vaccines. Thank you for the opportunity to review your work. I have made comments and suggestions that I hope will be helpful to this study.

Introduction: It would be helpful to readers to have a clearer understanding of the nature of vaccination service providers (VSPs) in community health centers in China. From your description, it appears that VSPs are a special unit that only provide vaccinations. Are patients referred to VSPs by other providers at the clinic? Do patients schedule appointments with VSPs directly?

You describe EPI and non-EPI vaccines. You indicate that EPI vaccinations are required for school and are free of charge. How much do the non-EPI vaccines cost?

Is there variable uptake of different non-EPI vaccines and are there variable costs of these vaccines? This would be helpful to know as cost seems to be an important determinant of patients’ decisions to get these vaccines.

Materials and Methods: (Study Design) You note that you used cluster sampling in your selection of participating health clinics in different provinces. Can you provide more information about the exact use of cluster sampling.

Results: (Quantitative Results) The logistic regression using one value as a referent. This works very well when there are two levels (eg., gender) but it does not as easily allow for comparisons among variables with three or more response categories.  For example, for Profession, (overall p value <0.01), nurses appear to be the outlying category.  Physicians and public health workers have similar distributions of recommendations of non-EPI vaccines. Likewise, there is a steady progression of recommending non-EPI vaccines with increased age of the VSP: <25 years and 25-35 years have very similar results and it’s difficult to determine where significant differences occur.

Reviewer 2 Report

I read this paper with great interest. I think that it merits publication. However, I have a few concerns that, to me, should be addressed.

I think that a mixed method approach is appropriate. However, I would expect the methodology to be better described, and more references should be added in this regard. 

For instance, I cannot understand how the survey has been built. This is a very sensitive topic, as I would expect the survey to be (mostly) gathered from/inspired by the current literature. 

See for instance (not to be cited, but as an example on how the survey has been designed/conceived):  Cobianchi, L. et al. (2021), “Team dynamics in emergency surgery teams: results from a first international survey”, World Journal of Emergency Surgery, Vol. 16, n. 47 https://doi.org/10.1186/s13017-021-00389-6 

The survey should be reported as an Appendix to the paper to stimulate other studies.

I would expect more details about the interviews.

Last but now least, I would stimulate the authors to dig deeper into the topic of knowledge translation in healthcare, as effective vaccine policies require adequate translation (what the authors call "communication"). 

I would highly reccomend the following references:

Dal Mas, F., Biancuzzi, H., Massaro, M. and Miceli, L. (2020), “Adopting a knowledge translation approach in healthcare co-production. A case study.”, Management Decision, Vol. 58 No. 9, pp. 1841–1862.

Dal Mas, F., Garcia-Perez, A., Sousa, M.J., Lopes da Costa, R. and Cobianchi, L. (2020), “Knowledge Translation in the Healthcare Sector. A Structured Literature Review”, Electronic Journal Of Knowledge Management, Vol. 18 No. 3, pp. 198–211.

Graham, I.D., Logan, J., M.B., H., Straus, S.E., Tetroe, J., Caswell, W. and Robinson, N. (2006), “Lost in knowledge translation: Time for a map?”, Journal of Continuing Education in the Health Professions, Vol. 26, pp. 13–24.

McAneney, H., McCann, J.F., Prior, L., Wilde, J. and Kee, F. (2010), “Translating evidence into practice: A shared priority in public health?”, Social Science and Medicine, Elsevier Ltd, Vol. 70 No. 10, pp. 1492–1500.

 Good luck in revising your work!

Author Response

Please see the attachment."
